# Effects of Co-Inoculation of Indole-3-Acetic Acid-Producing and -Degrading Bacterial Endophytes on Plant Growth

**Sabitri Adhikari Dhungana [1,2] and Kazuhito Itoh [1,3,*]** 

[1] The United Graduate School of Agricultural Sciences, Tottori University, 4-101 Koyama-minami, Tottori 680-8553, Japan; sabitriko@gmail.com
[2] Nepal Agricultural Research Council, Singhadurbar Plaza, Kathmandu 44600, Nepal
[3] Faculty of Life and Environmental Sciences, Shimane University, 1060 Nishikawatsu, Matsue, Shimane 690-8504, Japan
[*] Correspondence: itohkz@life.shimane-u.ac.jp; Tel.: +81-852-32-6521

**Abstract:** Bacterial production of indole-3-acetic acid (IAA) and its effects on plant growth have been frequently studied but there have been few studies on the ecology of IAA-degrading bacteria. In this study, among eight endophytic bacterial strains previously isolated from the same sweet potato sample including two IAA producers, *Klebsiella* sp. Sal 1 and *Enterobacter* sp. Sal 3, all of the strains showed IAA-degrading ability to some extent. *Herbaspirillum* sp. Sal 6 had the highest activity for IAA and tryptophan. When the IAA producers and the degrader were co-cultured in tryptophan-amended $N^+MR$ liquid medium, the concentrations of IAA decreased. Inoculation with *Klebsiella* sp. Sal 1, the highest IAA producer among the test strains, increased fresh root weight of tomato and radish, but the effect decreased by co-inoculation with IAA-degrading *Herbaspirillum* sp. Sal 6. Since both strains colonized plant parts at high populations, it was likely that the IAA degrader decreased IAA levels in the plants by degrading IAA and/or its precursor tryptophan. When IAA-producing biofertilizers are used, interactions with IAA degraders in plants should be considered.

**Keywords:** endophyte; indole-3-acetic acid (IAA); degradation; co-culture; co-inoculation; colonization

## 1. Introduction

Indole-3-acetic acid (IAA) is an important substance that regulates different developmental processes in plants such as cell division, elongation, and differentiation, as well as responses to gravity and light. The concentration of IAA in plants is crucial for controlling their growth [1], and it is controlled through biosynthesis, conjugation, degradation and intercellular transport [2]. IAA action on plant growth is concentration-dependent, and externally applied IAA showed an optimal concentration around $10^{-9}$ M for roots and $10^{-5}$ M for stems [1]. On the other hand, the application of IAA biosynthesis inhibitors resulted in a reduction of the endogenous IAA content followed by suppression of elongation and growth of tomato seedlings [3].

In addition to the endogenous IAA in plants, there has been a plethora of studies describing IAA-producing bacteria, including endophytes [4–7]. The application of IAA-producing bacteria to plants has shown significant increases in plant growth and yield as follows: *Pseudomonas fluorescens* and *Bacillus subtilis* in onion [8], *Rahnella aquatilis* in hybrid poplar [4], *Enterobacter ludwigii* in rice [9], and *Klebsiella pneumonia* in wheat and moth bean [10]. However, most of these studies were conducted under controlled conditions and single strain inoculation. It is well known that all plants are inhabited

internally by diverse microbial communities [11]; therefore, interactions in both positive and negative aspects would be expected among them.

In addition to IAA-producing bacteria, IAA-degrading bacteria have also been reported as members of the epiphytic community in pea plants [12] and in rhizospheres of pine trees [13] and tomato [14,15]. In this context, when IAA-producing bacteria are present as plant growth-promoting endophytes or are used as a biofertilizer, the presence of IAA-degrading bacteria in the endophytic community could eliminate or reduce the effects by decreasing the concentration of IAA in the plant. A number of studies have been carried out on IAA producers and their role in plant growth promotion, but the endophytic IAA-degraders have been studied less, despite their importance considering their potential interaction with IAA producers. In our previous work, eight endophytic bacterial strains isolated from the same sweet potato sample were studied for their IAA-producing and nitrogen-fixing abilities, and their potential for plant growth promotion [16]. The aim of this study was to examine the IAA-degrading ability of these strains, and to elucidate the effects of co-inoculation of the IAA producers with an IAA degrader on plant growth.

## 2. Materials and Methods

### 2.1. Bacterial Strains

Eight sweet potato bacterial endophytes used in our previous plant growth promotion study were used in this study [16] (Table 1).

**Table 1.** Sweet potato endophytic bacterial strains used in this study.

| Strain | * Most Similar Genus | Class | Accession Number | ** IAA-Producing Ability (µg/mL) |
|--------|----------------------|-------|------------------|----------------------------------|
| Sal 1 | *Klebsiella* sp. | Gammaproteobacteria | LC389410 | 65 |
| Sal 2 | *Flavobacterium* sp. | Flavobacteria | LC389415 | 0 |
| Sal 3 | *Enterobacter* sp. | Gammaproteobacteria | LC389433 | 40 |
| Sal 4 | *Rhizobium* sp. | Alphaproteobacteria | LC389434 | 20 |
| Sal 5 | *Stenotrophomonas* sp. | Gammaproteobacteria | LC389439 | 0 |
| Sal 6 | *Herbaspirillum* sp. | Betaproteobacteria | LC389442 | 0 |
| Sal 7 | *Agrobacterium* sp. | Alphaproteobacteria | LC389443 | 13 |
| Sal 8 | *Microbacterium* sp. | Actinobacteria | LC389445 | 4 |

* Most similar genus in the 16SrRNA gene sequence data base; ** Dhungana et al., 2018 [16].

### 2.2. IAA-degrading Ability of the Bacterial Strains

To determine IAA-degrading ability, the strains were cultivated in Modified Rannie (MR) [17] liquid medium amended with $NH_4NO_3$ at 0.1 g/L (N+MR) and 50 µg/mL IAA (FUJIFILM Wako Pure Chemical Corporation, Tokyo, Japan), and incubated at 26 °C at 150 rotations per minute (rpm) for 6 days. Control was set under the same conditions without inoculation. During the cultivation, optical density at 660 nm ($OD_{660}$) was monitored every day, and at 3 and 6 days, a 200 µL aliquot of the bacterial culture was taken and centrifuged at 10,000× $g$ for 10 min at 4 °C. The supernatant was mixed with a double volume of Salkowski reagent [18], and kept for 30 min in darkness. Then, the absorbance was measured at 530 nm using a UV-VIS spectrophotometer (UV-1700, Shimadzu, Kyoto, Japan).

### 2.3. Fate of IAA under Co-Cultivation of IAA-Producing and -Degrading Strains

The IAA-producing strains *Klebsiella* sp. Sal 1 and *Enterobacter* sp. Sal 3 were co-cultivated with the IAA-degrading strain *Herbasprillum* sp. Sal 6, and the fate of IAA was examined. Each strain was cultured in N⁺MR liquid medium for 2 days, washed twice with sterilized distilled water after centrifugation at 10,000× $g$ at 4 °C for 10 min., and then suspended to obtain equal populations at $10^9$ colony forming units (CFU)/mL. A 9 µL aliquot of the cell suspension was added to 3 mL of

N$^+$MR liquid medium amended with tryptophan, a precursor of IAA, at 200 μg/mL and incubated as mentioned above. OD$_{660}$, concentrations of IAA and tryptophan were measured at 12, 24, 36, 48, and 72 h after inoculation. Each strain was individually cultured under the same conditions. IAA and tryptophan in the culture were quantified using a Prominence Ultrafast Liquid Chromatography (UFLC) System equipped with a photodiode array detector (SPD-M20A) and a Shim-pack XR-ODS column (3.0 mm id × 100 mm, 2.2 μm) (Shimadzu, Kyoto, Japan). A solvent system of 0.5% formic acid and acetonitrile (75/25; vol/vol) was used for isocratic elution, and IAA and tryptophan were detected at 278 nm. The experiment was carried out in triplicate.

### 2.4. Effect of Co-Inoculation of IAA-Producing and -Degrading Strains on Plants

The IAA-producing (Sal 1) and -degrading (Sal 6) strains were individually inoculated and co-inoculated to tomato and radish plants, and their effects on the plants were examined. Seeds of tomato ('Momotaro' F$_1$ hybrid) and radish ('Taibyousoubutori') were purchased from Takii & Co., Ltd. (Kyoto, Japan), and surface sterilized by dipping in 70% ethanol for 1 min followed by 1% NaOCl for 13 and 18 min for tomato and radish, respectively, and washed 7–8 times with sterilized distilled water. The seeds were inoculated by dipping them overnight in the bacterial cell suspensions prepared as mentioned above. Controls were prepared by dipping the seeds in sterilized cell suspensions that had been autoclaved at 121 °C for 20 min. One inoculated ungerminated seed was sown in a glass tube (1.5 cm id × 10 cm) containing 1.5 g of sterilized vermiculite and 1 mL of liquid 1/2 MS medium [19] in which the amount of macroelement was adjusted to 1/2 strength, and capped with a silicon plug. Each treatment was conducted in 7–12 replications. Growth parameters were recorded after growing for 6 days in a phytotron (LH-240, Nippon Medical and Chemical Instruments Co., Ltd., Osaka, Japan) with 14 h light, 28/25 °C day/night temperature, and 6000 to 7000 lux light intensity in white fluorescent light conditions.

After recording the plant growth parameters, colonization of the inoculated strains was examined using one plant from each treatment. Rhizosphere colonization was examined by dipping and gently shaking the roots in sterilized distilled water. Colonization in the root and leaf was examined by macerating the separated parts in sterilized distilled water using a disposable homogenizer (BioMasher, Nippi, Tokyo, Japan) after surface washing with sterilized distilled water. An aliquot of the diluted samples was plated on N$^+$MR agar medium and the colonies that appeared were counted after 2 days of incubation at 26 °C. The morphologies of the colonies of the strains on the plate were clearly different, enabling separate counting.

### 2.5. Statistical Analysis

Statistical analysis was conducted using student's *t*-test or Tukey's test after one-way ANOVA using MINITAB ver. 14 (MINITAB Inc., USA).

## 3. Results

### 3.1. IAA-Degrading Activity of the Bacterial Strains

*Herbaspirillum* sp. Sal 6 degraded all the IAA in the medium within 3 days of cultivation (data not shown). The *Klebsiella* sp. Sal 1 and *Enterobacter* sp. Sal 3, IAA-producing strains significantly degraded IAA by ca. 40%, and *Rhizobium* sp. Sal 4, *Agrobacterium* sp. Sal 7 and *Microbacterium* sp. Sal 8 degraded IAA by ca. 20% at 6 days (Figure 1).

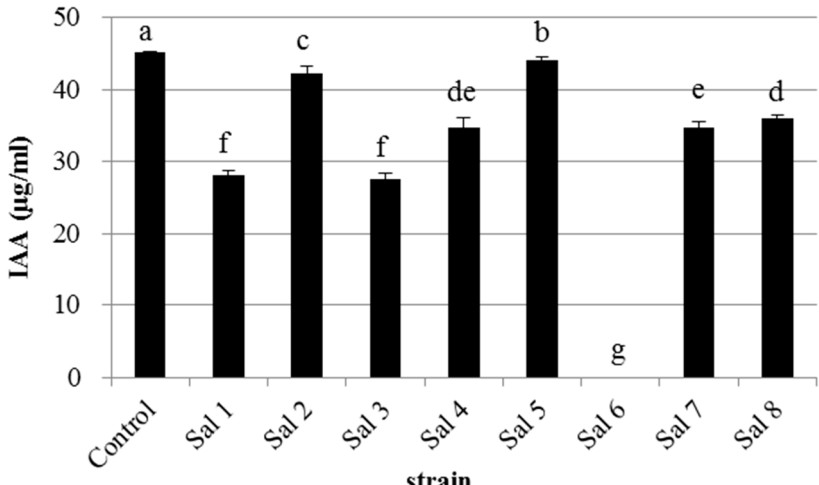

**Figure 1.** Degradation of indole-3-acetic acid (IAA) by sweet potato endophytes in $N^+MR$ medium. The bars represent standard deviation (n = 3) and different letters indicate significant differences at *P* < 0.01 by Tukey's test.

The IAA-degrading abilities of the strains were reflected in their growth differences in the media with and without IAA. The difference was the largest for the most IAA-degrading strain Sal 6, and it was larger in the moderately IAA-degrading strain Sal 3. In contrast, the difference in the other IAA-degrading strain Sal 1 was as same as the least IAA-degrading strain Sal 2 (Figure 2).

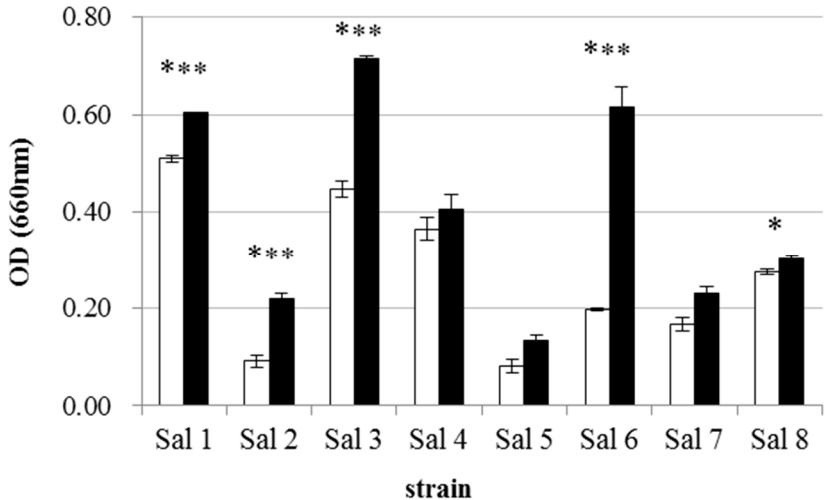

**Figure 2.** Growth of the endophytic bacterial strains in $N^+MR$ media at 6 days with (closed box) and without (open box) IAA. The bars represent standard deviation (n = 3) and asterisks indicate significant difference using student's *t*-test (*** for *P* < 0.001 and * for *P* < 0.05).

*3.2. Fate of IAA under Co-Cultivation of the IAA-producing and -degrading Strains*

The results for tryptophan and IAA are shown in Figure 3. Tryptophan was degraded by all strains, and IAA was produced by Sal 1 and Sal 3. When IAA-producing Sal 1 and Sal 3 were co-cultivated with IAA-degrading Sal 6, lower levels of IAA were detected in the culture, especially in Sal 1 + Sal 6.

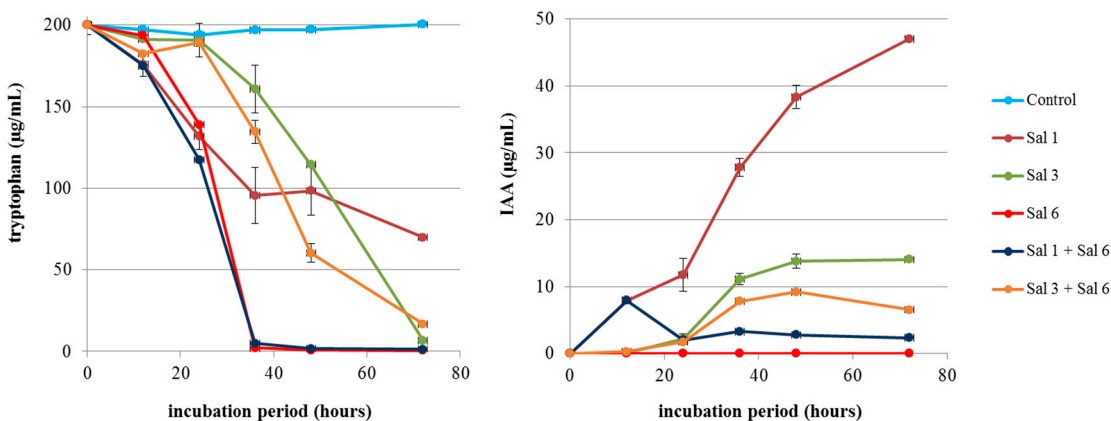

**Figure 3.** Fate of tryptophan and IAA under co-cultivation of the IAA-producing (Sal 1 and Sal 3) and -degrading (Sal 6) strains of sweet potato endophyte in tryptophan-amended medium. The bars represent the standard deviation (n = 3).

### 3.3. Effect of Inoculation of IAA-Producing and IAA-Degrading Strains on Plants

In tomato plants, inoculation with the IAA-producing strain Sal 1 caused a significantly higher fresh root weight than the control, but the effect was reduced by the co-inoculation of the IAA-degrading strain Sal 6 (Figure 4). The reduced level was as same as the individual inoculation of Sal 6. The root length and fresh plant weight were not affected by the inoculations.

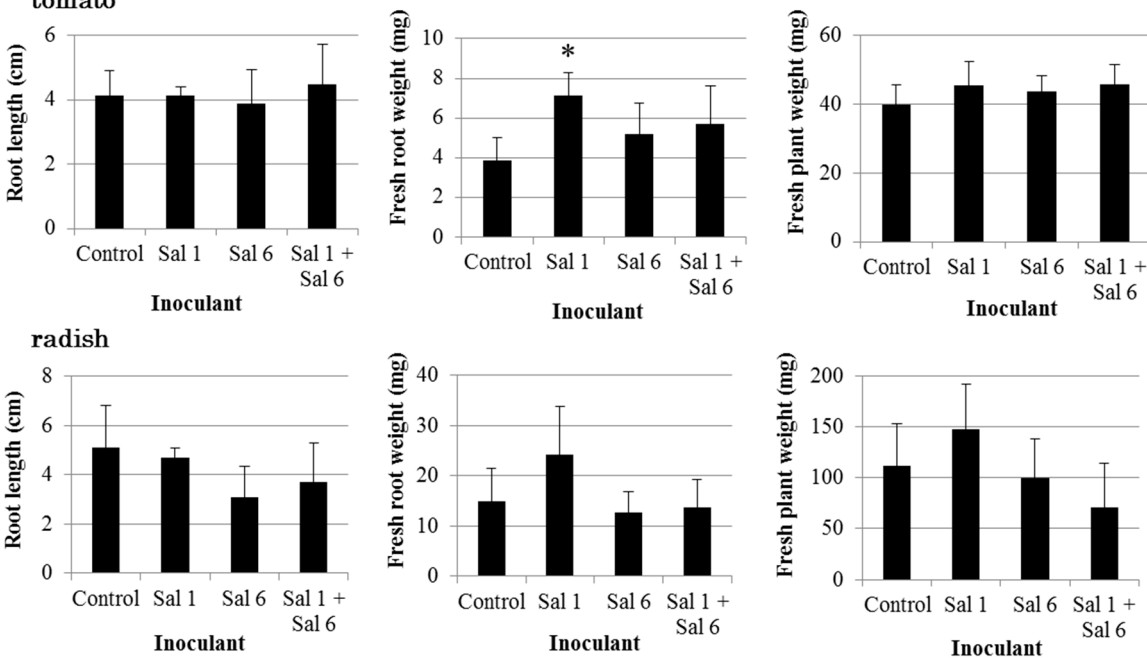

**Figure 4.** Effect of inoculation of IAA-producing (Sal 1) and -degrading (Sal 6) strains on the growth of tomato and radish plants. The bars represent the standard deviation (n = 7–12). Asterisk indicates a significant difference at $P < 0.001$ using Tukey's test.

In radish plants, a similar tendency was observed in the fresh root weight, but the difference was not significant due to the large deviations (Figure 4). In addition, the fresh plant weight was improved by the inoculation of IAA-producing strain Sal 1, and the effect was reduced by co-inoculation of the IAA-degrading strain Sal 6 compared to the individual inoculation of Sal 6, although apparent

differences were not statistically significant (Figure 4). In terms of root length, the inoculation of Sal 6 showed negative effects, but it was improved by the co-inoculation of Sal 1.

In the individual inoculations, both *Klebsiella* sp. Sal 1 and *Herbaspirillum* sp. Sal 6 colonized tomato and radish plants in high populations (Figure 5A). The colonization of the rhizosphere was 2–3 orders higher than the root and leaf in both species. The population of Sal 1 in tomato was 13 times higher in the root than in the leaf, whereas Sal 6 was seven times higher in the leaf than in the root. In the case of radish, the populations in the root and leaf were almost the same for both strains.

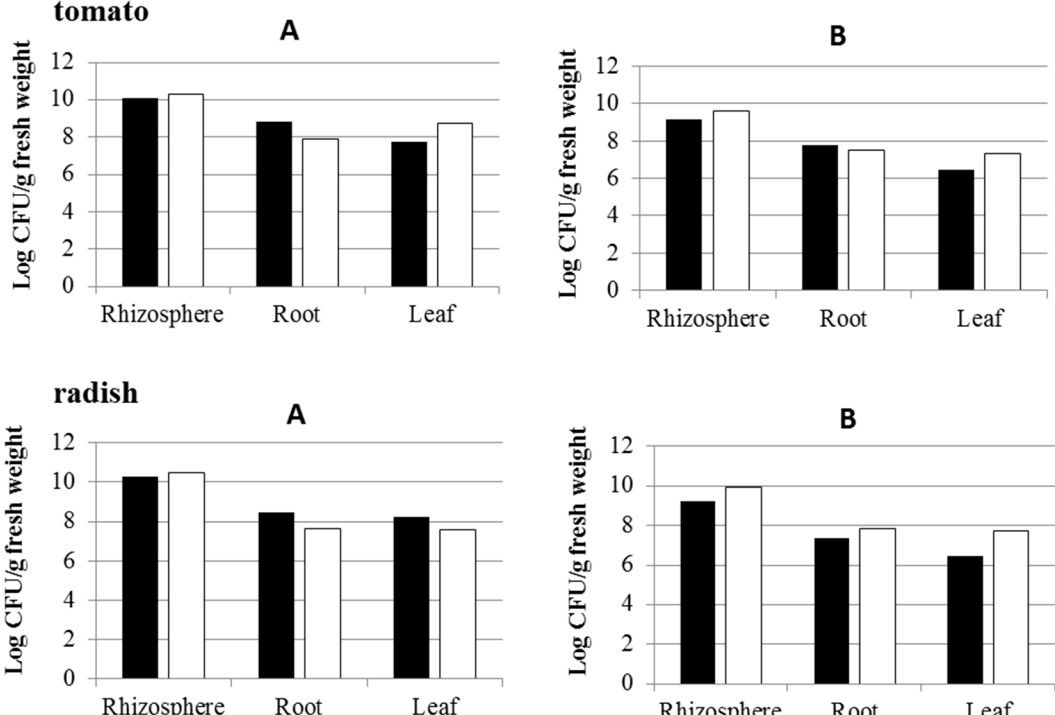

**Figure 5.** Colonization in plant parts by *Klebsiella* sp. Sal 1 (closed box) and by *Herbaspirillum* sp. Sal 6 (open box) individually (**A**) and co-inoculated (**B**) to tomato and radish seeds (n = 1).

In the co-inoculations, the rhizosphere, root and leaf were colonized by the bacteria similar to the individual inoculations (Figure 5B). The relative percentage of the population of Sal 6 was higher in all plant parts (75–95%) than that of Sal 1, except for the root of tomato (33%). The root colonization of Sal 1 was 23 and 8 times higher than in the leaf in tomato and radish, respectively. Higher root colonization was also observed in Sal 6, but the differences were smaller (1.4 and 1.3 times in tomato and radish, respectively). No colonies were observed for the control plants.

## 4. Discussion

All of the endophytic bacterial strains used in this study presented some ability to degrade IAA, with the non-IAA-producing *Herbaspirillum* sp. Sal 6 showing the highest activity. The high IAA-producing strains *Klebsiella* sp. Sal 1 and *Enterobacter* sp. Sal 3 also moderately degraded IAA. The growth difference between Sal 3 and Sal 6 with and without IAA in the medium suggested that the IAA-degraders utilized IAA as a source of energy. The use of IAA by the degraders as a carbon and energy source could be related to their colonization potential in the plants. The utilization of IAA as a sole source of carbon, nitrogen and energy for growth was also reported for *Pseudomonas putida* [20]. In our study, both IAA-producing and -degrading endophytes were found in the same sweet potato sample, suggesting that the concentration of IAA in plants would be affected by a balance of their activities.

Under in vitro conditions, the co-cultivation of IAA-degrading *Herbaspirillum* sp. Sal 6 decreased the concentration of IAA, which was produced by the co-cultured IAA producers. In addition, Sal 6 degraded tryptophan as well as IAA with the highest activity. Therefore, it was suggested that the level of IAA in the medium was reduced in two ways; one by degrading the produced IAA and another by degrading tryptophan, a precursor of IAA. There have been several examples of tryptophan-catabolizing bacteria utilizing tryptophan as the sole source of carbon and nitrogen, such as *Bacillus cereus*, *Pseudomonas aeruginosa*, *Ralstonia metallidurans* [21], *Pseudomonas aureofaciens* [22] and *Bacillus megaterium* [23]. In bacterial IAA biosynthesis, a tryptophan-independent IAA biosynthetic pathway [24] was also known in addition to the tryptophan-dependent pathways [25]; indole-3-pyruvate, indole acetamide, and tryptamine pathways. As a whole, bacterial metabolism of IAA, oxidative catabolism, IAA conjugation with sugar or amino acids, and the hydrolysis of such conjugates [26] should be considered as the determining factors of IAA levels in plants affected by endophytic bacterial communities. Therefore, complex interactions are expected among them that determine the IAA concentration. In the culture amended with tryptophan, the degradation rate of tryptophan by Sal 6 was reduced in the presence of Sal 3, suggesting a negative interaction between the two strains. There is also an example of positive interaction, such as *Sphingomonas* sp. SRS2, significantly enhancing the metabolism of phenylurea herbicide isoproturon by utilizing the methionine released by co-culturing an unidentified bacterial strain SRS1 [27]. Nutritional conditions have also been reported to affect bacterial IAA metabolism. IAA production decreased as the nitrogen levels increased in sweet potato endophytic strains *Klebsiella* sp. Sal 1 and *Enterobacter* sp. Sal 3 [16]. The degradation of IAA by epiphytic *Alcaligenes* and *Pseudomonas* strains was inhibited in the presence of glucose [12]. These results also suggested that the levels of IAA in the plant would be determined by the results of complex interactions among host plants, microbes and environmental conditions.

The inoculation by *Klebsiella* sp. Sal 1, which had the highest IAA-producing activity, improved lateral root growth and resulted in the significant increase of the fresh root weight of tomato, suggesting that IAA produced by the strain Sal 1 promoted plant growth. IAA regulates the promotion of lateral root growth [28], and inoculation of other IAA-producing plant-associated bacteria showed similar effects on inoculated strawberry [29] and mung bean [30]. Decreasing the effects by co-inoculation with IAA-degrading *Herbaspirillum* sp. Sal 6 also suggested that bacterial IAA improved plant growth. Since both IAA-producing and -degrading bacteria were found in all of the examined parts of the inoculated plants, the bacterial IAA produced seemed to be readily available to the co-existing IAA-degrading bacteria.

When IAA-producing strains are used as a biofertilizer in agriculture, their interactions with indigenous endophytic communities, especially with IAA-degrading endophytes, should be considered. They would interact with the inoculant positively and/or negatively, and the potential of the inoculants might be reduced when the degraders are active in the plant. An example of this was presented in this study, and this is the first such report to the best of our knowledge. The potential of an individual endophyte should be considered as a result of the interaction with its community. Therefore, IAA production and plant growth promotion observed in laboratory experiments would not act in the same way under actual growing conditions. These factors should be considered and the mechanisms of these microbial interactions should be studied further.

**Author Contributions:** S.A.D. and K.I. conceived and designed the experiments. S.A.D. performed the experiments, and interpreted and wrote the paper with significant contributions from K.I.

**Funding:** This research received no external funding.

**Conflicts of Interest:** The authors declare no conflict of interest.

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
