# Peer review of "Effects of Co-Inoculation of Indole-3-Acetic Acid-Producing and -Degrading Bacterial Endophytes on Plant Growth"

_horticulturae, doi:10.3390/horticulturae5010017_

Round 1
Reviewer 1 Report
The manuscript presented by Dhungana and Itoh aimed to show the potential beneficial effects of IAA-producing/degrading bacteria when inoculated on different agro-economically interesting plants. Interestingly, the authors highlighted the importance of the IAA-degraders in the interactions between plants and their associated bacteria. The balance between IAA-producers and IAA-degraders is very important in the design of effective biofertilizers.
The topic is very interesting and to the best of my knowledge, there are few papers discussing the effects of both IAA-producers and IAA-degraders. Actually, there is a plethora of studies describing the isolation and characterization of IAA-producing bacteria, most of them showing their effects under controlled conditions and single strain inoculation.
I believe that this study has a potential; however, there are some issues I think the authors must address before acceptance here or somewhere else.
The manuscript is written in a quite simple language. The introduction is too short and does not present enough background. Moreover, the discussion is vague, and the conclusions have not sufficient strength. I recommend the authors to add more studies related to the topic and emphasize the importance of the consideration of IAA-degraders for biofertilization schemes.
IAA and IAA-like molecules biosynthesis are a very common and extended capacity amongst members of the family Enterobacteriaceae. Also, there is a plethora of studies reporting strains belonging to genera Klebsiella and Enterobacter as excellent auxin producers. The authors assumed that strains selected in this study produce and degrade IAA. Please, indicate the provenience of the IAA used for the degradation assays. For IAA production, it is well known that Salkowski colorimetric technique is not specific of Indole-3-acetic acid (IAA); it can detect other IAA-like molecules, such as idoleacetamide and indolepyruvic acid (IPyA). Please, have this into account in order to present results and discuss them.
Please, provide reference for the IAA-degrading method. Do the authors germinated the seeds before place them onto tubes? Please, describe or give reference for the “dipping” method.
The number of replicates used to determine colonization is unacceptable. Please, provide explanation of why just one plant was used in this case and how they think this data contribute to the results. Moreover, I cannot see how results exposed in Figure 2 contribute to the study.
Why the authors do not perform statistical analyses for data presented in figure 5 and 6? What does data in figure 6 mean? There are no statistical analyses here, no replicates…
Minor comments:
Figures 3, 4 and 5 need editions to better understanding and also, the figure legend needs to be more informative.
Figure 4 and 5 should be merged.
Line 92. Fluorescent
Line 182-183. Please, provide reference.
Table 1. Replace Flavobacteriia by Flavobacteria
Author Response
Response to Reviewer 1
The manuscript is written in a quite simple language. The introduction is too short and does not present enough background. Moreover, the discussion is vague, and the conclusions have not sufficient strength. I recommend the authors to add more studies related to the topic and emphasize the importance of the consideration of IAA-degraders for biofertilization schemes.
We revised our manuscript according to the reviewer’s comment.
IAA and IAA-like molecules biosynthesis are a very common and extended capacity amongst members of the family Enterobacteriaceae. Also, there is a plethora of studies reporting strains belonging to genera Klebsiella and Enterobacter as excellent auxin producers. The authors assumed that strains selected in this study produce and degrade IAA. Please, indicate the provenience of the IAA used for the degradation assays. For IAA production, it is well known that Salkowski colorimetric technique is not specific of Indole-3-acetic acid (IAA); it can detect other IAA-like molecules, such as idoleacetamide and indolepyruvic acid (IPyA). Please, have this into account in order to present results and discuss them.
Source of IAA was presented in the revised manuscript. Salkowski technique was used only for the IAA degradation experiment (Fig. 1), in which the unspecific reaction would cause little problem. In the IAA production-degradation experiment (Fig. 3), UFLC was used for identifying tryptophan and IAA.
Please, provide reference for the IAA-degrading method. Do the authors germinated the seeds before place them onto tubes? Please, describe or give reference for the “dipping” method.
The reference for Salkowski colorimetric technique was presented in section 2.2 and there is no reference for UFLC analysis. As there is no reference for the dipping method, method of inoculation was revised to clarify the procedures.
The number of replicates used to determine colonization is unacceptable. Please, provide explanation of why just one plant was used in this case and how they think this data contribute to the results. Moreover, I cannot see how results exposed in Figure 2 contribute to the study.
We thought that one representative sample is enough to present the colonization of the inoculants in the plant. We did not expect statistical consideration. Similar levels of population was observed in the individually and co-inoculated samples of tomato and radish, therefore we think that the data seemed to be reliable.
Usage of IAA by the degraders as carbon and energy sources could be related to their colonization potential in the plants. It was incorporated in the discussion of the revised manuscript.
Why the authors do not perform statistical analyses for data presented in figure 5 and 6? What does data in figure 6 mean? There are no statistical analyses here, no replicates…
Statistical analysis was performed for Figure 5 but significant difference was not detected. For Figure 6, statistical analysis could not be performed due to analysis of one plant sample.
Figures 3, 4 and 5 need editions to better understanding and also, the figure legend needs to be more informative.
We would like to ask the reviewer what information is lacking in the figures.
Figure 4 and 5 should be merged.
We revised our manuscript according to the reviewer’s comment.
Line 92. Fluorescent
Spelling was corrected.
Line 182-183. Please, provide reference.
The sentence is followed by the representative reports with references.
Table 1. Replace Flavobacteriia by Flavobacteria
Flavobacteriia is the class of genus Flavobacterium.

Reviewer 2 Report
Dear Authors,
I have great honor to review manuscript entitled: “Effects of co-inoculation of indole-3-acetic acid (IAA) producing and degrading bacterial endophytes on plant growth”-Manuscript ID: horticulturae-424500, which is considered for publication in Horticulture Journal. This article presents very valid results with very interesting approach for analyses. However, authors did not avoid some errors. Therefore, I would like to present my reviewers comments to manuscript.
Introduction section:
This section is generally well written but is lack of some crucial information from literature about of IAA levels on sweet potato, tomato and radish because this plants are mainly connected with conducted by autors research. Authors must also present in more clearly way aim of the study. To do that authors should use sentence: “The aim of this study was….”
Materials and Methods section:
This paragraph is presented after introduction which is an error in manuscript construction. According journal publication rules this section must be after discussion and before conclusions. So current localization in manuscript is unacceptable. Because of that fact authors must also change/correct all numeration in text and in reference list in manuscript.
Table 1. Authors should deleted information **Dhungana et al., 2018 [14] in bottom of this table. Because this citation is made in main text of material and methods
rpm-should be explained when first used because it is not Si unit.
CFU- should be also explained clearly in main text
Authors add information about equipment used for analyses but they did not add any comments about software used for research did authors not used any?
Paragraph 2.5. Statistical Analysis has no information about used statistical software (version, manufacturer, country) what was confidence intervals?? Authors add information about intervals in results but all this information should be presented in materials and methods.
Results section:
Interesting however some quantitative data presented on Figures 3 and 6 has no statistical analyses which is very problematic to understand results correctly. Authors should analyzed this data by use statistical methods.
Because, all comments presented above I would like suggest to accepted this manuscript after minor revision.
Author Response
Response to Reviewer 2
Introduction section:
This section is generally well written but is lack of some crucial information from literature about of IAA levels on sweet potato, tomato and radish because this plants are mainly connected with conducted by authors research. Authors must also present in more clearly way aim of the study. To do that authors should use sentence: “The aim of this study was….”
We revised our manuscript according to the reviewer’s comment.
This paragraph is presented after introduction which is an error in manuscript construction. According journal publication rules this section must be after discussion and before conclusions. So current localization in manuscript is unacceptable. Because of that fact authors must also change/correct all numeration in text and in reference list in manuscript.
We prepared our manuscript according to the format of the latest published paper of this journal.
Table 1. Authors should deleted information **Dhungana et al., 2018 [14] in bottom of this table. Because this citation is made in main text of material and methods
The reference **Dhungana et al., 2018 [14] is for the asterisk data (**) presented in the same table.
rpm-should be explained when first used because it is not Si unit.
Full form of rpm was mentioned in the revised manuscript.
CFU- should be also explained clearly in main text
Full form of CFU was mentioned in the revised manuscript.
Authors add information about equipment used for analyses but they did not add any comments about software used for research did authors not used any?
Paragraph 2.5. Statistical Analysis has no information about used statistical software (version, manufacturer, country) what was confidence intervals?? Authors add information about intervals in results but all this information should be presented in materials and methods.
The software used was presented in the revised manuscript. We used standard deviation to express deviation of the data.
Interesting however some quantitative data presented on Figures 3 and 6 has no statistical analyses which is very problematic to understand results correctly. Authors should analyzed this data by use statistical methods.
For Figure 3, standard deviation of the data was presented in the figure.
For Figure 6, we thought that one representative sample is enough to present the colonization of the inoculants in the plant. We did not expect statistical consideration. Similar levels of population was observed in the individually and co-inoculated samples of tomato and radish, therefore we think that the data seemed to be reliable.

Round 2
Reviewer 1 Report
The manuscript was certainly improved.
Please, find my comments and point-by-point responses in the attached document.

Author Response
Response to Reviewer 1
R2. The introduction and discussion are fine now. I can assume if there are no other studies about this topic, you may not have what to compare. However, some of the claims go too far without any molecular data.
We are sorry, but we do not understand what “molecular data” means. If it means genetic information about degradation and biosynthesis of IAA, we think the information seems not to be critical for the subjects in our study. Considering the diversity of bacterial IAA metabolism, we added the information in discussion and considered it in relation to this study.
R2. I am not agreed about the claim that “unspecific reaction would cause little problem”. If you claim so, please, give reference.
When measuring formation of IAA from tryptophan, formation of reactive byproducts should be considered, but when measuring degradation of IAA, reduction of absorbance indicates disappearance of IAA and production of negligible amount of reactive byproducts.
R2. The reason given by the authors of not expecting statistical differences does not explain the lack of replicates. It is not acceptable and NOT a good practice to follow if you want to obtain reliable results.
Colonization of the inoculants in the test plants is important for presenting possible interaction between the IAA producing and degrading endophytes in the plants. Although we used only one of the replications, similar levels of population was observed in all of the six plants examined which include tomato and radish inoculated individually and co-inoculated. Therefore we think that their colonization in the plants seemed to be reliable and that this information is useful for readers.
R2. Authors did add information about number of replicates in Figure 3. Information about statistical method is missing in figure 5 (the authors claimed that the results are not significative, but to say so, the authors had to perform statistical analyses as well). In my opinion, these figures can be edited; for example, removing the word “inoculant” and leave it just once, add A, B, C… to recognize data easily… This is just a suggestion to improve the data presentation.
Statistical method was included as a result of merging the figures.
R2. Those figures should be merged as they represent same measured parameters for 2 different plants. As example, Figure 6 show the results of both plants.
Figures were merged according to the reviewer’s suggestion.
R2. Check the spelling. The correct name of this taxon is Flavobacteria, not Flavobacteriia.
Spelling was corrected.
